# Fabrication of Silk Resin with High Bending Properties by Hot-Pressing and Subsequent Hot-Rolling

**DOI:** 10.3390/ma13122716

**Published:** 2020-06-15

**Authors:** Hoang Anh Tuan, Shinji Hirai, Shota Inoue, Alharbi A. H. Mohammed, Shota Akioka, Tung Ngo Trinh

**Affiliations:** 1Research Center for Environmentally Friendly Materials Engineering, Muroran Institute of Technology, 27–1 Mizumoto, Muroran 050–8585, Japan; inoue-s@nihonseiko.co.jp (S.I.); mohammedahalharbi@gmail.com (A.A.H.M.); 18999809@mmm.muroran-it.ac.jp (S.A.); 2Technology and Alloys of casting Department, Research Institute of Technology for Machinery, Vietnam Engine and Agricultural Machinery Corporation-Joint Stock Company, No 25 Vu Ngoc Phan, Hanoi 100000, Vietnam; 3Function Polymers and Nano Materials Laboratory, Institute of Chemistry, Vietnam Academy of Science and Technology, No.08 Hoang Quoc Viet, Hanoi 100000, Vietnam; ngotrinhtung@gmail.com

**Keywords:** regenerated silk powder, resinification, hot-pressing, hot-rolling, Eri silk, Bombyx mori silk

## Abstract

This research reports the processability and mechanical properties of silk resins prepared by hot-pressing followed by hot-rolling and then analyzes their thermal and structural properties. The results show that regenerated silk (RS) resins are better suited for hot-rolling than Eri and Bombyx mori silk resins (untreated silk). When hot-rolling at 160 °C with a 50% of reduction ratio, maximum bending strength and Young’s modulus of RS resin reaches 192 MPa and 10.2 GPa, respectively, after pretreatment by immersion in 40 vol% ethanol, and 229 MPa and 12.5 GPa, respectively, after pretreatment by immersion in boiling water. Increased strength of the material is attributed to the increased content of aggregated strands and intramolecular linking of β sheets (attenuated total reflectance Fourier-transform infrared spectroscopy) and higher crystallinity (X-ray diffraction analysis). After hot-pressing and hot-rolling, RS resins have a stable decomposition temperature (297 °C).

## 1. Introduction

Over the centuries, silk threads have been fabricated conventionally into braided, knitted and non-woven matrices. Recently, it discovered that the material made from silk performed attractive properties such as biocompatibility, biodegradation, non-toxicity, and adsorption properties, which are important for medical applications [1,2]. Therefore, versatile processabilities from dissolved fibroin fibers to fabricate various morphologies of silk such as sponges, hydrogels, films, mats, micro-particles, and microneedles were made [2,3] for various medical applications such as tissue engineering, drug delivery [4,5,6,7], and osteotomy fixation [8,9]. On the other hand, after being dissolved, the fibroin structure has been found to be severely degraded in the peptide-strand structure of silk fibroin [10]. The correlations of the degraded phenomena on the experiment processability will be clarified in this paper.

Over the past two decades, significant research has focused on producing bio-resins (e.g., silk fibroin) via various methods as an environmentally friendly alternative for a wide range of accessories made from synthetic materials. These materials not only offer mechanical properties equal to the conventional plastics but are also biodegradable, biocompatible, and non-toxic, which are suitable for biomedical applications [11]. This is the reason why this silk-based resin is also a strong potential candidate to apply as biodegradable polymeric materials in biomedical fields, which is also mentioned in [12,13,14]. The goal is to attain mechanical properties similar to those of polyether ether ketone (PEEK) resin [15], which is one of the super engineering plastics that has been obtained. To reach this goal, the present research focuses on improving the mechanical properties of silk resins.

The conventional method to produce silk resin is by hot-pressing silk powder. Silk resin is produced by hot-pressing silk powder with 27% water between stainless plates under a pressure of 44 MPa and at a temperature of 160 °C for 1 h [16]. Silk resin may also be produced by hot-pressing laminated silk fibroin sheets [17]. However, the water absorbance of the laminated sheets significantly exceeds that of turtle shell. The hardness is equivalent to that of tortoiseshell, which can serve as a substitute material for turtle shells in applications such as eyeglass frames and ornaments. However, the bending strength of silk resin is 70 MPa, which is much less than that of tortoiseshell (225–333 MPa).

In previous work, we fabricated silk resin with a bending strength of 100 MPa and a bending Young’s modulus of 4.5 GPa [18] by hot-pressing commercial silk powder obtained by the dissolution of Bombyx mori (B. mori) silk thread after degumming in an aqueous solution of a neutral salt, desalination, and then freeze-drying [19]. We used commercial silk powder with 20 wt% of water added, heated to 150–170 °C using spark plasma sintering followed by cold-pressing at 20–30 MPa for a short time. The resulting silk resin has a thermal conductivity of 0.44 W/(mK) and a glass transition temperature of 180 °C.

Recently, we introduced a more straightforward method to prepare silk resin that involves hot-pressing silk powders directly by milling raw silk fibers (without degumming) from B. mori and Eri silk fibers at 170 °C and 31.2 MPa [20]. The maximum three-point bending strength and the Young’s modulus of the resulting silk resin are 122 MPa and 8.7 GPa, respectively, for B. mori silk resin and 100 MPa and 8.5 GPa, respectively, for Eri silk resin. Analysis by attenuated total reflectance Fourier-transform infrared (ATR-FTIR) spectroscopy shows that, after resinification and drying, the β-sheet content in B. mori silk increases and the random coil (RC) structure decreases. In contrast, after resinification, the secondary structure in Eri silk resin goes from a random coil structure to a β-strand structure, which converts to a β-sheet structure after drying.

Another approach is to use hot-rolling to fabricate silk resin sheets from silk powder, which allows hot-rolling equipment to be used for continuous production [21]. Silk powder mixed with 20 wt% of water is wrapped in a pulp sheet and processed by hot-rolling at 130 °C. A curing area of 90% is attained with a rolling reduction ratio of 60%. Because the bending strength decreases significantly upon increasing the rolling reduction ratio, the optimum three-point bending strength of the silk resin sheet is 100 MPa for a 40% rolling reduction ratio. According to previous X-ray diffraction (XRD) analyses [18,20,21,22], the silk resin sheets have the silk II crystal structure, which is the β-sheet crystal. Casting from an aqueous solution of silk fibroin produces a β-rich silk film that depends on the casting conditions, such as drying temperature, drying rate, type of substrate, and the initial concentration of fibroin [23,24,25]. Another method of producing the β-sheet form is by using temperature-controlled water vapor annealing [26,27,28] or by soaking in a polar solvent such as ethanol [29,30,31,32,33,34,35,36,37].

The goal of the present work is to prepare high-strength silk resin sheets by hot-rolling and to investigate the processability of these sheets with and without a pretreatment consisting of soaking in ethanol or boiling water. We also evaluate the mechanical, thermal, and structural properties of the silk resin sheets. This approach is motivated by the hypothesis that preparing silk resin sheets by hot-pressing and subsequent hot-rolling will improve the mechanical properties of the sheets because the hot-pressing process increases the silk II crystal structure, which is then aligned by hot-rolling the silk resin sheets. To optimize the hot-rolling process, the original resins underwent several pretreatments, in particular, soaking in ethanol or boiling water. A previous study reported that the presence of water reduces the strength of the crystalline structure by reducing the number of hydrogen bonds, the hydrogen bond lifetime, and the peak rupture force while increasing the specific interaction energy [37]. The same research also confirmed that this effect of moisture and the position-dependent mechanical response of silk II crystallites should help enhance the mechanical properties of silk fibroin [38].

## 2. Materials and Experiment

### 2.1. Materials

This study used commercially available silk powder (KB Seiren, Ltd., Shiga, Japan) derived from B. mori silk produced by dissolving the waste of spinning-silk thread in an aqueous neutral salt solution, followed by desalination, solidification, precipitation, dehydration, drying, and finally pulverization [19]. The result is called “regenerated silk” (RS) powder. Pulverized powders of B. mori silk thread or Eri silk thread without degumming were prepared by milling as previously reported [20]. Ethanol (Kanto Chemical. Co., Inc., Tokyo, Japan, 99.5%) and distilled water were used for the pretreatment. Figure 1 shows a flow chart describing how to prepare the silk resin.

### 2.2. Amino Acid Composition

The amino acid content in the fibroin samples was determined using a High-Speed Amino AL-8800 instrument (Hitachi, Japan), which hydrolyzes the peptide bonds of the silk-fibroin powder and resins in a 4 mol/L methanesulfonic acid solution, allowing automated amino acid analysis (Wako Pure Chemical, Osaka, Japan) with 0.2 wt% 3-(2-aminoethyl) indole used as acid catalyst at 110 °C for 24 h.

### 2.3. Resinification

Silk powder, after being filled into a 20-mm-diameter cylindrical stainless-steel die, was pressed at 31.2 MPa while being heated from room temperature to 170 °C using a hot-pressing apparatus (H300–05, AS One, Osaka, Japan) in order to sinter the silk resin. All the process mentioned above is called resinification. After the resinification process, the silk resin was removed immediately from the stainless-steel die and cooled to room temperature. The resinification method to obtain silk resin is described in detail elsewhere [20].

### 2.4. Ethanol and Boiling-Water Pretreatments

The hot-pressed silk resin was next pretreated by immersion in 40–100 vol% ethanol for 20–30 min or by immersion in boiling deionized water (100 °C) for 60 min.

### 2.5. Hot-Rolling

The pretreated silk resin was calendered using manually operated hot-rolling equipment (IMC-1989, Imoto Machinery Co. Ltd., Kyoto, Japan) with roller dimensions of φ60 mm × 132 mm and operated at 150–170 °C.

The rolling reduction ratio R ranged from 0% to 50%, where R is the percent change in thickness due to rolling per unit initial thickness:(1)$$R=\frac{h_{1}-h_{2}}{h_{1}}\times 100\%$$

In Equation (1), *h*_1_ is the initial sample thickness before rolling, and *h*_2_ is the final sample thickness after rolling, as shown in Figure 2.

The goal of the hot-rolling process is to increase the content of crystalline structure (β sheets, aggregated strands) in silk fibroin and to improve the alignment, thereby increasing the strength of the resin. As mentioned above, water is an important factor in the hot-rolling process, which means that efficient pretreatments should produce a significant concentration of hydrophilic amino acids. Moreover, the highest-strength hot-rolled resin samples are those with minimal surface defects, such as wavy edges, zipper cracks, edge cracks, alligatoring, etc.

### 2.6. Scanning Electron Microscope

Scanning electron micrographs of the surface morphology were obtained by first carbon coating the samples with a CC-40F carbon coater (Meiwafosis Co. Ltd. Shinjuku, Japan) and then imaging using a JSM-6510 scanning electron microscope (SEM; Jeol, Japan) at magnifications of ×200 and ×500.

### 2.7. Mechanical Properties

Three-point bending tests were implemented using an Autograph AGS-X series universal tensile testing machine (Shimadzu, Tokyo, Japan) with the silk resin cut into 3 × 18 × 2.5 mm^3^ samples. The tensile strength of all samples was measured both parallel to the rolling direction and normal to the surface of the resin sheet (i.e., orthogonal to the rolling direction) on a support interval of 14 mm and at a stretching speed of 0.5 mm/min^−1^. The results reported are the average of a minimum of five samples produced under identical hot-rolling conditions. The bending elastic modulus was calculated from the stress-strain curve. Depending on the type of pretreatment (ethanol or water boiling), RS resins were dried prior to testing at 100 °C in a vacuum-type oven for 4 to 12 h to vary the water content and thereby determine how water content affects the mechanical properties of RS resins. The water content of the silk samples was measured using Karl Fischer titration (860 KF Thermoprep, Metrohm AG, Herisau, Switzerland).

### 2.8. Thermogravimetric Analysis TGA/DTG

Thermogravimetric analysis and difference thermo gravimetry (TGA and DTG) were implemented using a TG/DTA7300 instrument (Hitachi, Tokyo, Japan) to determine the thermal behavior of RS silk powders, the original RS resins, and RS hot-rolled resins by measuring the percent weight loss and decomposition peaks from the TGA and DTG curves. The measurements were made from 30 to 600 °C at a heating rate of 10 °C/min under a nitrogen atmosphere (50 mL/min). The test samples weighed 8–10 mg.

### 2.9. Structural Analyses

The RS resins were cut into 10 × 10 mm^2^ samples for structural analysis using ATR-FTIR (FT/IR-6600, Jasco, Tokyo, Japan) over the range 700–4000 cm^−1^ and with a resolution of 4 cm^−1^. Second derivatives and multipeak Gaussian fits were used to determine the secondary structural composition of amide I. In addition, the crystallinity was determined using XRD (Ultimate IV Protectus, Rigaku, Tokyo, Japan) with Cu Kα radiation (40 kV and 40 A) at a scan speed of 0.2 deg/min and with a sidestep 0.01 deg over the 2θ range from 5° to 35°.

## 3. Results and Discussion

### 3.1. Processability

Figure 2 shows a schematic representation of the rolling process. The resin was drawn into the rolling gap by the frictional force between the surface of the material and the rollers and then deformed under the compressive force of the roller. To draw the resin through the gap between the rollers, the coefficient of friction must satisfy
(2)$$Fcos\alpha =\mu P_{R}cos\alpha \;\geq P_{R}sin\alpha \Leftrightarrow \mu \geq \;tan\alpha$$
where *μ* is the coefficient of static friction, α is the bite angle, *P*_R_ is the rolling pressure, and *F* is friction shear stress.

Figure 3 shows photographs of the three types of silk resins after various pretreatments (including original resin, ethanol treatment, and boiling treatment) and those resins after hot-rolling with the maximum rolling reduction ratios. The treatment of all silk resins is shown through the pictures in the second row. After that, the treated silk resins were hot-rolled in several reduction ratios (5%, 10%, 20%, 30%, 40%, and 50%). The pictures in the first row present the hot-rolled resins with the maximum reduction ratio. The maximum rolling reduction ratio, which was 50%, was used for the resin derived from RS powder. However, for the resins derived from the pulverized powders of B. mori silk and Eri silk threads, the hot-rolling process is ineffective, even at the lower reduction ratio, i.e., 10% (see Figure 3). This result may be explained by the fact that Young’s modulus of resin derived from B. mori silk and Eri silk (8.7 and 8 GPa, respectively [20]) significantly exceeds that of resin derived from RS powder (4.5 GPa [18]). Therefore, silk resin made from RS powder is more easily processed by hot-rolling than resin made from B. mori or Eri silk powders, which have massive chains of silk fibroin. This phenomenon is due to the strong silk-fibroin peptide-strand structure of these two resins that prevented the B. mori silk resin and Eri silk resin from the hot-rolling process. Moreover, a similar observation reported in [10] indicates that regenerated silk derived from degumming and dissolution, the process of which was similar to that used herein to obtain RS powders, severely degrades the peptide-strand structure of silk fibroin. Because of the degraded peptide-strand structure, the RS resin was softened, which is the most important condition supported for the hot-rolling process.

As shown in Table 1, the hydrophobic/hydrophilic ratio of the amino acid composition of RS powder, B. mori silk powder, and Eri silk powder follows the order RS powder < B. mori silk powder < Eri silk powder. This result relates to the capability of the resin to hold free water in the fibroin structure. The more hydrophilic amino acids in RS compared with B. mori silk and Eri silk leads to greater water absorbance in the fibroin structure of the former. This evidence explains why RS resin is suitable for hot-rolling, especially after the boiling-water pretreatment, whereas the B. mori silk and Eri silk are not suitable for hot-rolling.

Thus, we focus hereinafter on RS: all results mentioned below (SEM, mechanical properties, TGA-DTG, ATR-FTIR, and XRD) are thus for RS. In this study, all RS resins could be hot-rolled. Additionally, upon increasing the rolling reduction ratio, crack defects appeared at the edge of the resin. Such defects are due to the fact that the sample edges were not pressed by the rollers and spread freely; however, the sample was still drawn in the rolling direction, which explains why the sample was torn by the rolling force and the secondary tensile stress [39].

### 3.2. Scanning Electron Microscopy (SEM)

Figure 4 shows SEM images of resins derived from RS powder before and after hot-rolling and with different pretreatments and a 50% reduction ratio. Generally, RS hot-rolled resins without pretreatment and with 100 vol% ethanol pretreatment have a rough morphology, with big cracks, whereas the RS hot-rolled resins pretreated with 40 vol% ethanol pretreatment and with boiling-water pretreatment have a smoother morphology. Upon increasing the SEM magnification, small fractures are detected in the RS hot-rolled resins pretreated with 40 vol% ethanol; however, at the same magnification, no such fractures are detected in the original RS resin or in the RS hot-rolled resin with boiling-water pretreatment. This result indicates that cracks at the surface of hot-rolled resins degrade the mechanical properties of the material.

### 3.3. Mechanical Properties

#### 3.3.1. Effect of Pretreatment on Mechanical Properties of Hot-Rolled Resins

Table 2 shows mechanical properties (three-point bending strength, Young’s modulus and strains) of the original resin, the hot-rolled resin without pretreatment, the hot-rolled resin pretreated using 40 vol% ethanol, and the hot-rolled resin pretreated using boiling water, with all resins prepared from RS powder. The water content of all samples was fixed at 2%–4%.

Overall, all of the RS hot-rolled resins have higher bending strength and Young’s modulus than the original silk resin. However, the mechanical properties of the RS hot-rolled resins are determined by several factors: the rolling reduction ratio, the orientation of the sample (parallel versus orthogonal to the rolling direction), the type of pretreatment, the water content before and after hot-rolling, and the ethanol concentration (for ethanol pretreated RS resins).

First, the bending strength and Young’s modulus increase upon increasing the rolling reduction ratio. The bending strength and Young’s modulus of hot-rolled resins with 50% rolling reduction ratio and with samples parallel to the rolling direction are superior to those of hot-rolled resins with 30% rolling reduction ratio. Furthermore, the bending strength and Young’s modulus orthogonal to the rolling direction for the hot-rolled resins are less than those parallel to the rolling direction for the same resins. This result suggests that hot-rolling affects mainly the structural orientation in the rolling direction by improving the silk II crystal structure and its orientation.

The strain is independent of hot-rolling conditions such as pretreatments, deformation ratio, and the hot-rolling direction.

For RS hot-rolled resin pretreated with boiling water (40 vol% ethanol), the largest bending strength and Young’s modulus are 229 MPa and 12.5 GPa (192 MPa and 10.2 GPa), respectively. The mechanical properties of the RS hot-rolled resin pretreated with ethanol are inferior to those of the RS hot-rolled resin pretreated with boiling water. This result is attributed to the different effects of ethanol and water molecules on the fibroin structure. The different effects of ethanol versus water molecules on the fibroin structure are discussed in the following sections.

#### 3.3.2. Effect of Concentration of Ethanol Pretreatment on the Mechanical Properties of Hot-Rolled Resins

Table 3 shows how the concentration of the ethanol pretreatment affects the bending strength and Young’s modulus of RS hot-rolled resins when the rolling reduction ratio and drying times are fixed at 30% and 50% and 4–6 h, respectively. For a rolling reduction ratio of 30%, the three-point bending strength and elastic modulus are independent of the ethanol concentration. Conversely, for 50% rolling reduction, the mechanical properties are maximal at 40 vol% ethanol concentration but gradually decrease with increasing ethanol concentration. In particular, resin soaked in 100 vol% ethanol has inferior mechanical properties compared with resin soaked in 40 vol% ethanol. This result is attributed to the polarity of the ethanol molecule, which allows it to easily hydrate to water, following which the water is separated from the molecular chain upon dehydration in the ethanol solvent. Therefore, pretreatment with a high-concentration ethanol solution decreases the water content of the original RS resin by dehydration. Conversely, ethanol pretreatment reportedly increases the β-sheet content of silk fibroin [29,30,31,32,33,34,35,36,37], which is consistent with the improved mechanical properties of the RS resin. The increase of the mechanical properties caused the appearance of the crack on the RS hot-rolled resins. As a result, the reduced mechanical properties of the resins are attributed to the cracks in the samples.

For the 40–80 vol% ethanol pretreatment, the mechanical properties of hot-rolled resin increase slightly because of the presence of water in the solvent. After ethanol pretreatment, the resin is covered by water, which supports the hot-rolling process (how water affects the fibroin structure is discussed in the next section). Comparing SEM images shows that the reduced mechanical properties of the RS hot-rolled resin correspond to the development of cracks on the resin surface, which in turn corresponds to the increase in ethanol concentration.

#### 3.3.3. Effect of Water Content on Mechanical Properties of Hot-Rolled Resins Pretreated with Boiling and with 50% Reduction Ratio

Figure 5 shows the three-point bending strength as a function of strain for RS hot-rolled resins with a rolling reduction ratio of 50% and for different water contents. The results show that a long drying time increases the bending strength and the Young’s modulus but decreases the strain by decreasing the water content. The initial water content of the RS hot-rolled resin is in the range 10–20 wt%, which is considered to be freezing water; this result explains somewhat the breaking of the hydrogen bond between the molecules of amorphous silk, which is why the resin with high water content is easier to hot-roll than the resin with low water content.

The RS hot-rolled resin dried for about 4–6 h has a water content between 7 to 8 wt%, including free water and bound water. A water content of 8.69 wt% is considered the minimum required for the resin to be plasticized for the hot-rolling process. The plasticity of the silk resins follows the curve through the elastic region, where the bending strength of the brittle resins (i.e., resins with less than 6.67 wt% water content) is not displayed. Notably, the 2.64 wt% water content delivers the highest bending strength and Young’s modulus for the RS hot-rolled resin with bound water, which was produced by hot-rolling at 229 MPa after drying for 12 h. However, drying for over 12 h dramatically reduces the water content to 0.26 wt%, which decreases the bending strength, Young’s modulus, and strain of the RS hot-rolled resin because of the loss of the non-freezing bound water, which contributes to cross linking in the polymeric molecule. Previous studies reported a similar mechanism for water molecules in RS films [40,41,42].

In conclusion, the optimal water content is 2%–4%, which leads to the best mechanical properties for RS hot-rolled resin.

### 3.4. Thermal Analysis

The thermal properties of all RS samples were obtained using TGA and DTG measurements, the results of which are shown in Figure 6. In brief, the TGA curves of all RS resins have a similar shape. For the RS powder, the initial weight loss (about 9 wt%) at approximately 70 °C corresponds to the evaporation of free water from the RS powder [14,43,44]. This weight loss is not detected in the TGA or DTG curves for the RS resins, which means that the silk resin with the best mechanical properties contains no free water. Nevertheless, the first-stage decomposition peaks of the RS resin appear at 229, 230, and 226 °C for RS resin, hot-rolled RS resin without boiling-water pretreatment, and RS resin with boiling-water pretreatment, respectively. These peaks are attributed to the removal of bound frozen water from the silk fibroin [14]. The second stage of weight loss for all silk resins appears at 297 °C, whereas that of the RS powder is at 301 °C, which corresponds to protein pyrolysis, namely, the progressive deamination, decarboxylation, and depolymerization arising from breaking (poly)peptide bonds, and the concomitant carbonization of the primary structure [43,44,45,46,47]. Finally, the similar TGA and DTG spectra for all silk resins reflect the stability of the decomposition temperature of ~297 °C.

### 3.5. Attenuated Total Reflectance Fourier-Transform Infrared Spectroscopy

Peptide chains, which are the structural repeat units of proteins, have up to nine characteristic infrared-active bands (called amide A, B, I, II, …, VII). The amide I, II, and III bands are major bands in the protein infrared spectrum and can be assigned to various secondary structures. However, the amide III band is of limited use for extracting structural information because it is a complex band resulting from a mixture of coordinate displacements. The amide I band (between 1600 and 1700 cm^−1^) is assigned mainly to C=O stretching vibrations of the backbone conformation (70%–85%), whereas the amide II band is primarily assigned to the combination of N–H in-plane bending vibrations (40%–60%) and C–N stretching vibrations (18%–40%) [48,49]. The amide III band falls in the range 1510–1580 cm^−1^ [48,49].

#### 3.5.1. Effect of Pretreatment on Regenerated Silk Resin

Table 4 shows the ATR-FTIR assignment for the RS resins before and after ethanol pretreatment (40 vol% and 100 vol%) and boiling-water pretreatment. The ratio of the intensity of the amide I to amide II bands (see Table 4) rises in the following order: RS resin with 100 vol% ethanol pretreatment < original RS resin < RS resin with 40 vol% ethanol pretreatment < RS resin with boiling-water pretreatment. For the RS resins with pretreatment and subsequent hot-rolling, a higher amide I/II ratio correlates with better mechanical properties of the RS hot-rolled resins, which means that RS resin with pretreatment and a higher amide I intensity is better served by hot-rolling than RS resin with no pretreatment and a lower amide I intensity.

#### 3.5.2. Effect of Hot-Rolling on Regenerated Silk Resin with and without Boiling-Water Pretreatment

Figure 7 compares the RS samples (including RS powder, RS resin, and RS hot-rolled resin with and without boiling-water pretreatment). The highest absorbing amide I peaks appear at 1627, 1621, 1620, and 1618 cm^−1^, respectively, which correspond to β sheets [50,51]. Conversely, the highest absorbing amide II and amide III peaks for RS powder appear at 1516 and 1230 cm^−1^, respectively. The highest absorbing amide II and amide III peaks for all RS resins (original resin and hot-rolled resin with and without boiling-water pretreatment) are redshifted to 1515 cm^−1^ for amide II and to 1229 cm^−1^ for amide III, which are assigned to β sheets [30,49,52].

The assignments of each band are given in previous works [50,51] as follows: 1605–1615 cm^−1^ corresponds to aggregated strands, 1616–1627 cm^−1^ corresponds to intermolecular β sheets, 1628–1637 cm^−1^ corresponds to intramolecular β sheets, 1638–1655 cm^−1^ corresponds to random coils, 1656–1662 cm^−1^ corresponds to α helices, 1663–1696 cm^−1^ corresponds to β turns, and 1697–1703 cm^−1^ corresponds to antiparallel β sheets. The percent content of the various secondary structural components is given in Table 5, and the resulting spectra for all samples were calculated and are shown in Figure 8. Resinification converts the random coil and β-turn structure to the aggregated-strands structure. In addition, the intermolecular β-sheet links are actually reconstructed intramolecular β-sheet links. Hot-rolling leads to a secondary structure of resin that is similar to that obtained by resinification, albeit with smaller ratios of random coil and β-turn to aggregated strands and of intermolecular to intramolecular β-sheet links. In addition, the calculated secondary structure implies that a higher content of aggregated strands and intramolecular β-sheet links gives better mechanical properties. More details on the relationship between the secondary structure of fibroin and the mechanical properties are available in the literature [3].

### 3.6. X-ray Diffraction

Based on an XRD analysis, the structure of silk fibroin is classified into two primary crystalline structures: silk I and silk II. The silk I conformation corresponds to the metastable crystalline structure that was originally found in spinning dope [53,54,55]. The silk II conformation is the stable structure with a well-oriented β sheet [3,53,54,55,56,57]. The mechanism by which the silk I conformation changes to silk II (the structure that fibroin takes after spinning) before spinning by the silkworm remains unclear.

Figure 9 shows the XRD spectra, and Table 6 lists the separation of all peaks. The peak intensities of silk II increase in the following order: RS powder < original RS resin < RS hot-rolled resin without boiling-water pretreatment < RS hot-rolled resin with boiling-water pretreatment. In addition, the peak position after resinification shifts to higher angles whereas that for hot-rolling shifts to lower angles. The percent crystallinity of the silk samples increases in the following order: RS powder 29.3% < original RS resin 38.8% < RS hot-rolled resin without boiling-water pretreatment 54.4% < RS hot-rolled resin with boiling-water pretreatment 57.7%. Additionally, peak 2 of the RS powder indicates that the lattice spacing of 4.6 Å decreases to 4.4 Å, which corresponds to silk I after resinification [58]. After hot-rolling, lattice spacing increases to 4.7 Å, which corresponds to silk II [58]. A similar explanation holds for peak 3, with a lattice spacing of 4.3 Å (silk II) for the RS powder, decreasing after hot-rolling to 3.1 Å (silk I) for RS resin [58].

In a nutshell, the hot-pressing process converts sintered silk powder to silk resin. The water content in the silk powder evaporates during hot-pressing, which helps to adhere the bulk silk via hydrogen bonds. Moreover, the ATR-FTIR results show that the random coil and β-turn structure (silk I) of silk-fibroin transform into the aggregated strand structure (silk II). However, the aggregated strand structure (silk II or crystallized structure) slightly increases after hot-rolling. The XRD results indicate that the percentage of silk II increases after hot-pressing and hot-rolling, which is similar to the results obtained using ATR-FTIR, as mentioned above. Furthermore, the silk II structure continues increasing from 38.8% to 54.4% for RS hot-rolled resin without pretreatment and to 57.7% for RS hot-rolled resin with boiling-water pretreatment. This result means that, during hot-rolling, silk II is oriented in the direction of the drawing or shearing force. A similar mechanism is invoked in previous studies to explain this phenomenon [59,60,61,62,63]. In the present study, the mechanical properties of silk resin are improved by increasing the percent of material oriented in the direction of crystallization.

## 4. Conclusions

The mechanical properties of silk resins increase upon hot-rolling the resins. It is easier to hot-roll RS resin than Bombyx mori silk resin or Eri silk resin because of the inferior mechanical properties of the former. To support the hot-rolling process, we subjected the resin samples to an ethanol pretreatment or a boiling-water pretreatment to degrade the mechanical properties of the resins before hot-rolling. The results show that the RS hot-rolled resin with boiling-water pretreatment has the best mechanical properties (229 MPa three-point bending strength and 12.5 GPa Young’s modulus).

The mechanical properties of RS hot-rolled resin depend on the water content. Lower water content in the resin corresponds to better mechanical properties. The best mechanical properties are attained with 2.64% water content. However, the mechanical properties decline dramatically when the water content decreases below 2.64%, which we attribute to non-freezing bound water, which plays a role in cross-linking. Removing this water breaks the polymeric structure of the resin.

Compared with the other research [15], the highest bending strength of RS hot-rolled resin (229 MPa) is stronger than those of PEEK (~170 MPa), which is one of the super engineering plastics obtained. Moreover, Young’s modulus of RS hot-rolled resin (12.5 GPa) is four times greater than Young’s modulus of PEEK (~3 GPa) [15]. The pretreatment and hot-rolling process not only improved the mechanical properties of RS resin but also enhanced the water stability and the stability during the in vitro degradation test of the silk materials because of the increase of the silk fibroin crystal conformation [5]. According to this, the functions of silk resins have been further extended based on the advantageous properties of silk materials (biocompatibility, biodegradation, non-toxicity, high strength, etc.). It is predicted that the silk bio-resin will be used as a biodegradable polymeric material in plentiful applications that play an important role for biomedical applications (e.g., biodegradation orthopedic implants).

## Figures and Tables

**Figure 1 materials-13-02716-f001:**
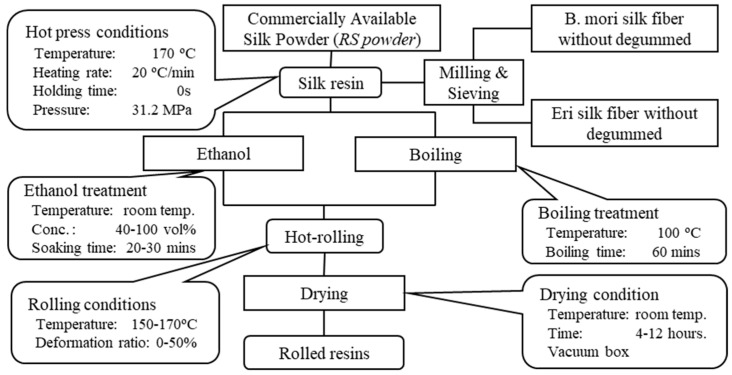
The flow diagram for preparing the hot-rolled silk resins.

**Figure 2 materials-13-02716-f002:**
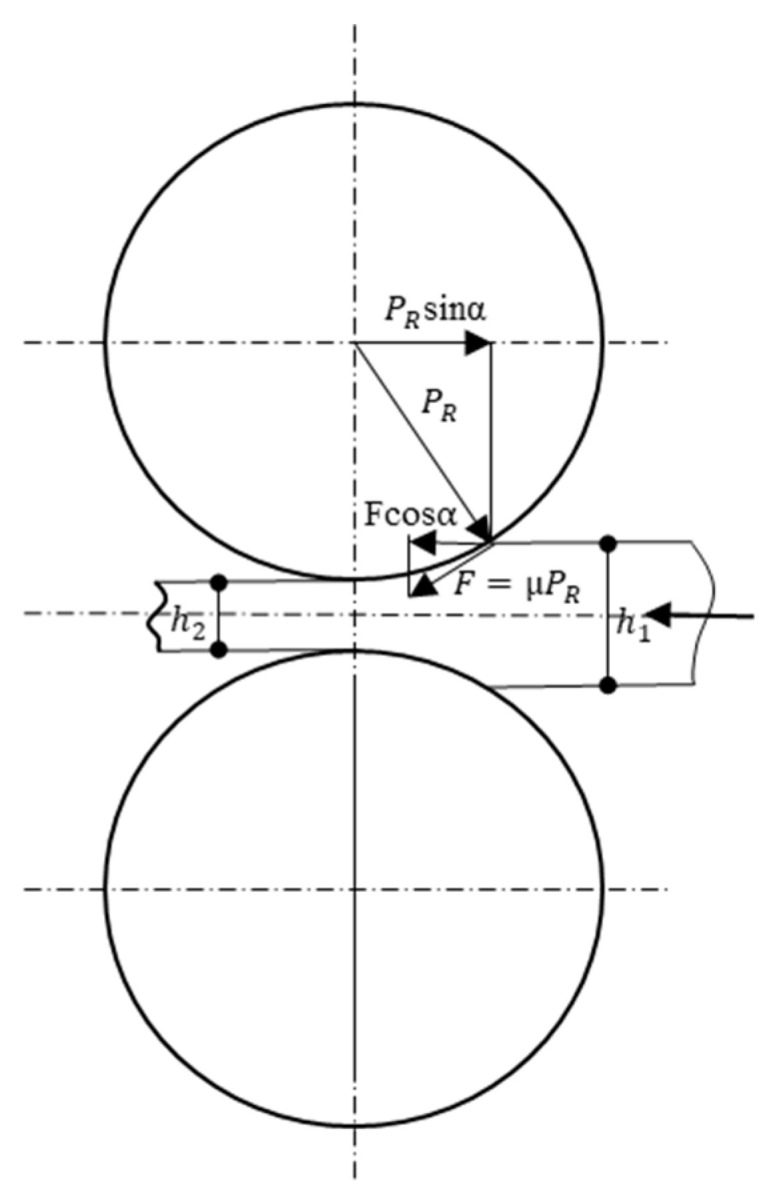
Schematic representation of the rolling process.

**Figure 3 materials-13-02716-f003:**
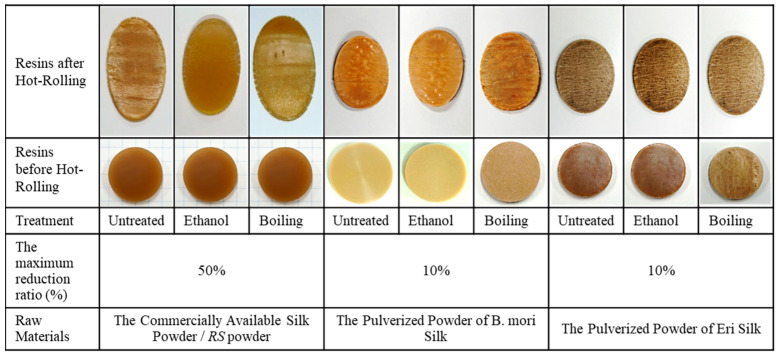
The appearance of the silk resins before and after the hot-rolling. The rolling reduction ratio indicates the maximum rolling reduction ratio here.

**Figure 4 materials-13-02716-f004:**
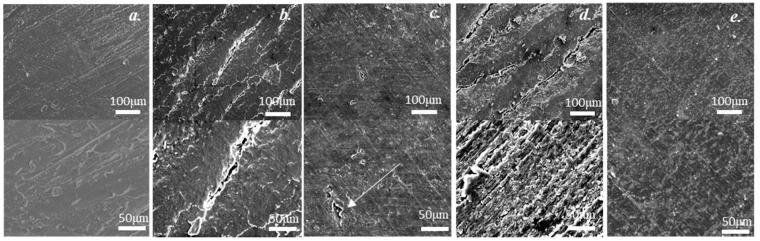
Scanning electron microscope (SEM) images of the RS original resin and the RS hot-rolled resins with the different treatments and 50% reduction ratio: (**a**) the original RS resin; (**b**) the RS hot-rolled resin without treatment; (**c**) the RS hot-rolled resin with 40 vol% ethanol treatment; (**d**) The RS hot-rolled resin with 100 vol% ethanol treatment; (**e**) the RS hot-rolled resin with boiling treatment.

**Figure 5 materials-13-02716-f005:**
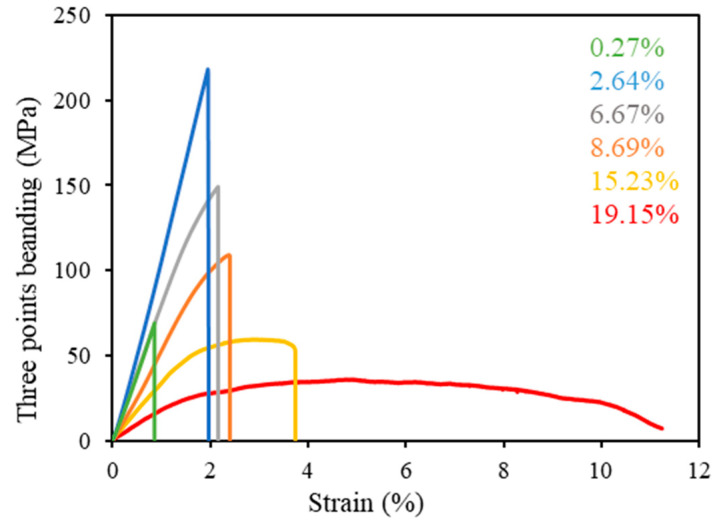
The three-point bending curve of RS hot-rolled resins (50% reduction ratio) with boiling treatment in the rolling direction.

**Figure 6 materials-13-02716-f006:**
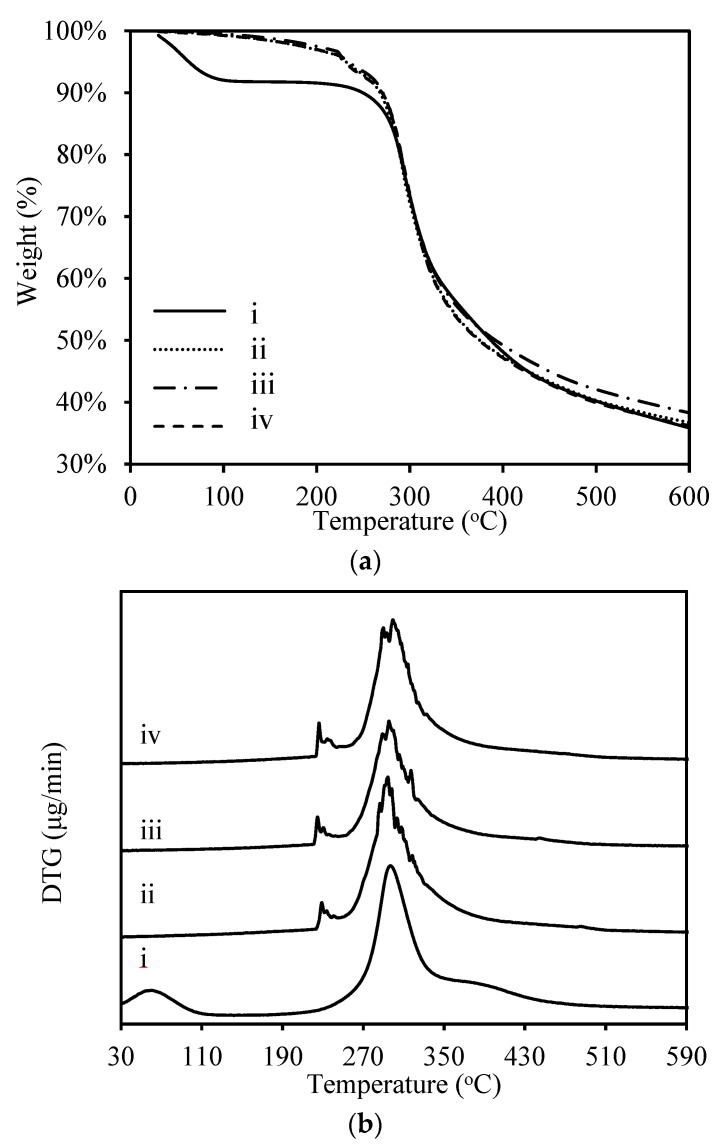
Thermogravimetric analysis (TGA) (**a**) and difference thermo gravimetry (DTG) (**b**) of the RS powder, the original RS resins, and the RS hot-rolled resin with 50% reduction ratio, (i. RS powder; ii. RS original resin; iii. RS hot-rolled resin without treatment; iv. RS hot-rolled resin with boiling treatment).

**Figure 7 materials-13-02716-f007:**
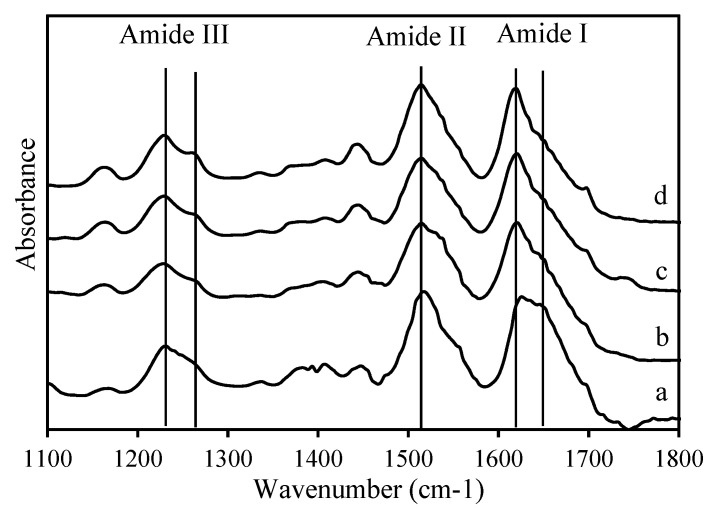
**ATR**-FTIR spectra of the RS powder, resin, and rolled resin: (a) the RS powder, (b) the original RS resin, (c) the RS hot-rolled resin without the boiling treatment, (d) the RS hot-rolled resin with the boiling treatment.

**Figure 8 materials-13-02716-f008:**
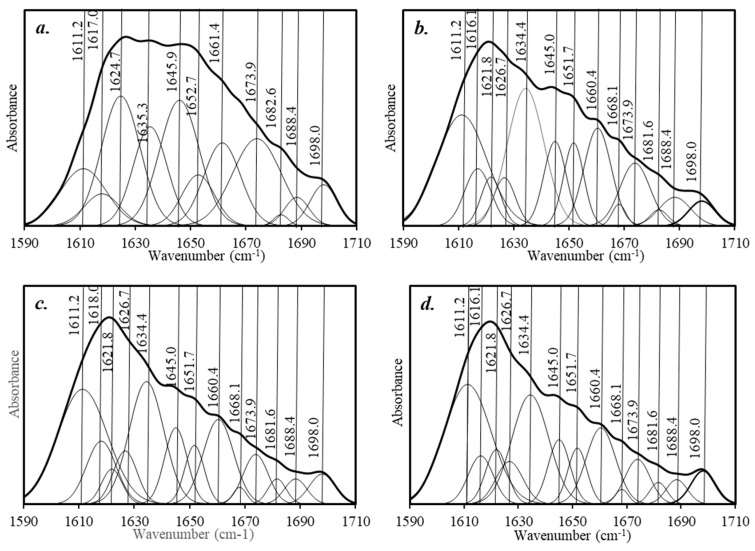
The ATR-FTIR spectra and the curve fitting of the ATR-FTIR spectra to determine the secondary of amide I: (**a**) the RS powder, (**b**) the original RS resin, (**c**) the RS hot-rolled resin without the boiling treatment, (**d**) the RS hot-rolled resin with the boiling treatment.

**Figure 9 materials-13-02716-f009:**
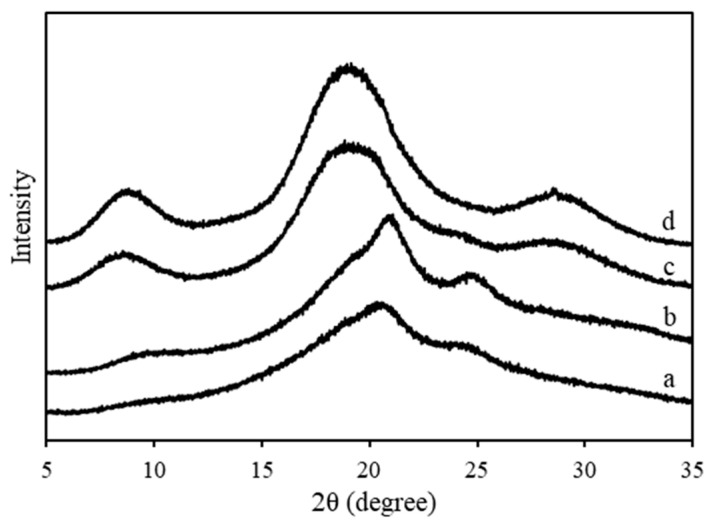
XRD spectra of (a) the RS powders, (b) the RS resins, (c) the RS hot-rolled resin without the boiling treatment, and (d) the RS hot-rolled resin with the boiling treatment.

**Table 1 materials-13-02716-t001:** Amino acid composition of B. mori silk, Eri silk, and regenerated silk (RS).

Amino Acid		Reference [20]		Current Study
		B. mori Silk	Eri Silk	RS
Hydrophilic	Pro	0.56	0.42	0.53
	Gly	38.50	31.44	42.75
	Ser	14.48	6.98	10.51
	Glu	1.92	0.92	1.35
	Asp	4.81	4.12	1.82
	Cys	0.17	0.08	0.11
	Thr	2.55	0.52	0.98
	Lys	0.79	0.27	0.28
	His	0.47	1.26	0.19
	Arg	1.07	1.97	0.52
Hydrophobic	Ala	25.30	44.75	30.61
	Met	0.08	0.01	0.04
	Val	2.62	0.60	2.35
	Phe	0.67	0.23	0.76
	Ile	0.69	0.38	0.68
	Leu	0.68	0.37	0.54
	Tyr	4.59	5.63	5.68
Hydrophobic/hydrophilic ratio		2.23	3.59	0.69
Basic amino acid (mol%)		2.33	3.5	0.99
Acidic amino acid (mol%)		6.73	5.04	3.17
Basic/acidic ratio		0.35	0.69	1.3

**Table 2 materials-13-02716-t002:** The mechanical properties of the original RS resin samples and the hot-rolled silk resins in different rolling reduction ratios without and with different pretreatments. (The water content of all samples is fixed at around 2%–4%).

Samples			Mechanical Properties (Average Values)		
			Bending Strength (MPa)	Modulus (GPa)	Strain (%)
Original resin		Rolling reduction ratio	93 ± 14	4.7 ± 0.5	1.8 ± 0.2
Orthogonal of rolling direction (for all hot-rolled resin)		30%	95 ± 16	6.1 ± 0.5	1.8 ± 0.2
		50%	108 ± 12	6.5 ± 0.6	1.8 ± 0.2
Rolling direction	Hot-rolled resins without treatment	30%	94 ± 14	6.2 ± 0.4	1.8 ± 0.2
		50%	152 ± 28	10.2 ± 0.4	1.8 ± 0.2
	Hot-rolled resins with ethanol treatment (40 vol%)	30%	122 ± 18	7.2 ± 0.3	1.8 ± 0.2
		50%	175 ± 17	9.7 ± 0.5	1.8 ± 0.2
	Hot-rolled resins with boiling treatment	30%	127 ± 14	7.8 ± 0.4	1.8 ± 0.2
		50%	220 ± 9	12.2 ± 0.3	1.8 ± 0.2

**Table 3 materials-13-02716-t003:** Mechanical properties of hot-rolled RS resin with different ethanol concentration treatment and reduction ratios (30%, 50%).

Reduction Ratio		30%		50%	
		Bending Strength (MPa)	Modulus (GPa)	Bending Strength (MPa)	Modulus (GPa)
Ethanol concentration	40%	126 ± 16	6.9 ± 0.3	175 ± 17	9.7 ± 0.5
	60%	122 ± 18	7.2 ± 0.3	178 ± 9	9.7 ± 0.5
	80%	127 ± 15	6.9 ± 0.4	167 ± 12	9.4 ± 0.9
	100%	125 ± 7	7 ± 0.2	124 ± 4	8.3 ± 0.7

**Table 4 materials-13-02716-t004:** Attenuated Total Reflectance Fourier-transform infrared (ATR-FTIR) assignment of the original RS resin and RS resin after different treatment conditions before the hot-rolling process.

Sample	Amide I		Amide II		Intensity Ratio of Amide I/II
	(a)	(b)	(a)	(b)	
Original Resin	1620 ± 0.9	1	1513 ± 0.9	0.795 ± 0.02	1.257 ± 0.02
40 vol% Ethanol treatment	1619 ± 0.9	1	1514 ± 0.9	0.762 ± 0.05	1.312 ± 0.05
100 vol% Ethanol treatment	1620 ± 0.9	1	1513 ± 0.9	0.812 ± 0.04	1.231 ± 0.04
Boiling treatment	1619 ± 0.9	1	1517 ± 0.9	0.749 ± 0.02	1.335 ± 0.02

(a) is the frequency position/wavenumber (cm^−1^) of amide I and amide II peaks. (b) is the intensity of the peaks after normalization.

**Table 5 materials-13-02716-t005:** The secondary structure content (%) of amide I of RS powder, original RS resin, and the RS hot-rolled resin without and with boiling treatment.

Sample	α-Helix	β-Sheet			β-Strand	Random Coil	β-Turn	Aggregated Strands
		a	b	Antiparallel				
Powder	10.81 ± 0.52	17.82 ± 0.91	11.99 ± 0.22	3.95 ± 0.13	4.11 ± 0.21	23.78 ± 1.24	18.24 ± 0.82	9.31 ± 0.34
Original Resin	10.88 ± 0.43	8.35 ± 0.51	20.71 ± 1.03	2.72 ± 0.21	6.22 ± 0.34	14.65 ± 0.97	13.99 ± 0.74	22.48 ± 1.64
Hot-rolled resin without boiling treatment	11.43 ± 0.64	8.78 ± 0.49	18.76 ± 0.63	3.49 ± 0.39	8.34 ± 0.57	12.76 ± 0.92	11.19 ± 0.86	25.25 ± 1.42
Hot-rolled resin with boiling treatment	10.74 ± 0.55	11.17 ± 0.53	19.23 ± 1.11	3.82 ± 0.18	5.75 ± 0.41	11.89 ± 1.02	10.68 ± 0.68	26.71 ± 1.66

a: Intermolecular β-sheets. b: Intramolecular β-sheets.

**Table 6 materials-13-02716-t006:** Analysis peaks of XRD results of RS samples and their crystallinity percentage.

Sample	Powder		Original Resin		Hot-Rolled Resin without Boiling Treatment		Hot-Rolled Resin with Boiling Treatment	
	Position	% area	Position	% area	Position	% area	Position	% area
Peak 1	8.2 ± 0.2	1.8 ± 0.1	9.6 ± 0.4	1.7 ± 0.1	8.6 ± 0.3	7.2 ± 0.2	8.7 ± 0.2	8.0 ± 0.2
Peak 2	19.4 ± 0.1	21.4 ± 0.3	20.1 ± 0.2	28.4 ± 0.4	18.8 ± 0.2	39.5 ± 0.5	18.8 ± 0.4	37.9 ± 0.4
Peak 3	20.6 ± 0.1	3.5 ± 0.1	20.8 ± 0.1	3.8 ± 0.1	20.5 ± 0.1	0.5 ± 0.1	20.3 ± 0.1	1.1 ± 0.1
Peak 4	24.4 ± 0.3	1.6 ± 0.1	24.9 ± 0.2	4.3 ± 0.2	23.2 ± 0.3	1.6 ± 0.2	21.8 ± 0.3	3.8 ± 0.1
Peak 5	32.2 ± 0.2	1.0 ± 0.1	32.7 ± 0.2	0.5 ± 0.1	29.1 ± 0.1	5.5 ± 0.3	29.1 ± 0.3	6.8 ± 0.2
Total	29.3 ± 0.7%		38.8 ± 0.9%		54.4 ± 1.3%		57.7 ± 1.0%	

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
