# Peer review of "Fabrication of Silk Resin with High Bending Properties by Hot-Pressing and Subsequent Hot-Rolling"

_materials, 2020, doi:10.3390/ma13122716_

Round 1

Reviewer 1 Report

The authors design the experiments and try to clarify the possible implications of the silk resin manufacturing method with the processability and mechanical properties of the material obtained.
This paper may have some value as a cautionary tale for material experimental design but the title is a bit misleading: the application of these specific materials for biodegradable bone implants is only a hypothesis. While they can be used as biodegradable bone implants, there is no indication of how this happens and there is no discussion about it. Please modifie the title according to the hypothesis of the study or add a biodegradability study according to a bone system.

Author Response

 Thank you for inviting us to submit a revised draft of our manuscript entitled “Fabrication of Silk Resin with High Bending Properties by Hot-Pressing and Subsequent Hot-Rolling” to Special Issue "Silk-Based Biomaterials" of Materials. We also appreciate the time and effort you and each of the reviewers have dedicated to providing insightful feedback on ways to improve our paper. Thus, it is with great pleasure that we resubmit our article for further consideration. We have made changes to reflect the detailed suggestions that you graciously made, and we hope that our edits and the responses provided below address satisfactorily all the issues and concerns that you and the reviewers have noted.

Reviewer 1

The authors design the experiments and try to clarify the possible implications of the silk resin manufacturing method with the processability and mechanical properties of the material obtained.

This paper may have some value as a cautionary tale for material experimental design but the title is a bit misleading: the application of these specific materials for biodegradable bone implants is only a hypothesis. While they can be used as biodegradable bone implants, there is no indication of how this happens and there is no discussion about it. Please modifie the title according to the hypothesis of the study or add a biodegradability study according to a bone system.

Response: We agree with your suggestion. The biodegradable bone implants application is just a hypothesis; our data is not enough to affirm this. However, we have continued to study this kind of material on in vivo, in vitro experiments for testing the biocompatibility, biodegradation timeline in vivo, etc. to clarify this. The title of the article has already removed the “applied for Biodegradable Bone Implants” part. The introduction and the conclusion about this also has been rewritten in line 45 to line 48 and line 432 to line 434.

Reviewer 2 Report

This is an interesting study by Tuan et al comparing the mechanical properties of hot rolled silk resins from different source of silk (regenerated silk, Eri or Bombyx mori) and treatments (ethanol vs boiling).  Overall regenerated silk was found to be superior.

My main comment is that both the title and abstract mention that the materials can be used for biodegradable bone implants however no data is shown to support these claims. I suggest removing this from both unless a significant amount of data is presented to shown that hot press silk resin has been tested as a bone implant material. Furthermore simply the biocompatibility and strength of the material are not an indicator at all of its potential.

Given that the paper compares regenerated silk to raw silk, the introduction would benefit from a brief explanation about how this process is carried out and what are there differences in the composition of both.

It would also be helpful to explain what a silk resin is and how this would be used in different applications such as biomedical.

In the results - please explain the rolling reduction % in relation to Figure 2.

The photgraphs in Figure 3 need some additional explaination to assist the reader to understand.

It would be better to carry out your own amino acid composition analysis for B mori silk and Eri silk rather than comparing to reference (especially an unexamined patent..)

In general the paper is well written. The formatting is poor and this needs to be addressed before publication.

Author Response

 Thank you for inviting us to submit a revised draft of our manuscript entitled “Fabrication of Silk Resin with High Bending Properties by Hot-Pressing and Subsequent Hot-Rolling” to Special Issue "Silk-Based Biomaterials" of Materials. We also appreciate the time and effort you and each of the reviewers have dedicated to providing insightful feedback on ways to improve our paper. Thus, it is with great pleasure that we resubmit our article for further consideration. We have made changes to reflect the detailed suggestions that you graciously made, and we hope that our edits and the responses provided below address satisfactorily all the issues and concerns that you and the reviewers have noted.

Reviewer 2

This is an interesting study by Tuan et al comparing the mechanical properties of hot rolled silk resins from different source of silk (regenerated silk, Eri or Bombyx mori) and treatments (ethanol vs boiling).  Overall regenerated silk was found to be superior.

My main comment is that both the title and abstract mention that the materials can be used for biodegradable bone implants however no data is shown to support these claims. I suggest removing this from both unless a significant amount of data is presented to shown that hot press silk resin has been tested as a bone implant material. Furthermore, simply the biocompatibility and strength of the material are not an indicator at all of its potential.

Response: We agree with your suggestion. The biodegradable bone implants application is just a hypothesis; our data is not enough to affirm this. However, we have continued to study this kind of material on in vivo, in vitro experiments for testing the biocompatibility, biodegradation timeline in vivo, etc. to clarify this. The title of the article has already removed the “applied for Biodegradable Bone Implants” part. The introduction and the conclusion about this also has been rewritten in line 45 to line 48 and line 432 to line 434.

Given that the paper compares regenerated silk to raw silk, the introduction would benefit from a brief explanation about how this process is carried out and what are there differences in the composition of both.

It would also be helpful to explain what a silk resin is and how this would be used in different applications such as biomedical.

Response: We agree with you and have incorporated this suggestion in our paper. The introduction has been already added a brief explanation about the dissolved process inline 33 to line 40

In the results - please explain the rolling reduction % in relation to Figure 2.

Response: We have already explained about the rolling reduction ratio in the manuscript. Please search in section 2.5. Hot-rolling in line 128 to line 132.

The photographs in Figure 3 need some additional explanation to assist the reader to understand.

Response: We agree with your suggestion. The paragraph explained in Figure 3 was rewritten inline 180 to line 198.

It would be better to carry out your own amino acid composition analysis for B mori silk and Eri silk rather than comparing to reference (especially an unexamined patent.)

Response: The reference about amino acid composition analysis for B mori silk and Eri silk is our previous research which already published. Hence, this result is our own result. Please find out the information in the article named “Preparation of silk resins by hot pressing Bombyx mori and Eri silk powders”

In general, the paper is well written. The formatting is poor and this needs to be addressed before publication.

Response: Thank you for your suggestion. The formatting of the manuscript has already fixed.

Round 2

Reviewer 2 Report

While I am pleased that the authors have removed references to bone transplants in their paper - I think suggesting that silk will replace plastic is somewhat naive.

They also mention that the goal of the work is to attained similar mechanical properties to polyether ketone - but no comparison is made in the results or discussion.

I suggest avoiding any specific applications but rather talk about the benefits of silk materials e.g. biocompatibility, degradable.

While some effort has been made to address my other concerns especially regarding regenerated silk vs raw silk and explanations about Figure 3. These need to be improved prior to publication.

Author Response

Dear Editor and Reviewers,

We are so happy to have an opportunity to revise our manuscript now entitled “Fabrication of Silk Resin with High Bending Properties by Hot-Pressing and Subsequent Hot-Rolling” (771903). In this revised manuscript, we have carefully considered all reviewers’ comments and suggestions. As being instructed, we have attempted to succinctly explain changes made in reaction to all comments. We reply to each comment in a point-by-point fashion. After addressing the issues raised, we believe that the quality of the paper is much improved.

Reviewer 2

While I am pleased that the authors have removed references to bone transplants in their paper - I think suggesting that silk will replace plastic is somewhat naive.

Response: We have been checked all according to your ideas and totally agreed with your insightful suggestions. All of the applications which were mentioned in the abstract (inline 27), introduction (line 44 - 45) and conclusion (line 429) were removed. Moreover, the reference about PEEK has been already added.

They also mention that the goal of the work is to attain similar mechanical properties to polyether ketone - but no comparison is made in the results or discussion.

I suggest avoiding any specific applications but rather talk about the benefits of silk materials e.g. biocompatibility, degradable.

Response: Thank you very much for your recommendations. The comparison between RS hot-rolled resin and poly-ether ether ketone (PEEK) was added in the conclusion inline 429 to line 435. This conclusion also mentioned the benefits of silk materials e.g. biocompatibility, degradable.

While some effort has been made to address my other concerns especially regarding regenerated silk vs raw silk and explanations about Figure 3. These need to be improved prior to publication.

Response: We’d like to sincerely appreciate your insightful comments as well as your suggestions to complete the paper.